# Randomized Controlled Immunotherapy Clinical Trials for GBM Challenged

**DOI:** 10.3390/cancers13010032

**Published:** 2020-12-24

**Authors:** Stefaan W. Van Gool, Jennifer Makalowski, Simon Fiore, Tobias Sprenger, Lothar Prix, Volker Schirrmacher, Wilfried Stuecker

**Affiliations:** 1Immun-Onkologisches Zentrum Köln, Hohenstaufenring 30-32, 50674 Koln, Germany; makalowski@iozk.de (J.M.); fiore@iozk.de (S.F.); tobias@sprenger-praxis.de (T.S.); v.schirrmacher@web.de (V.S.); stuecker@iozk.de (W.S.); 2Biofocus, Berghäuser Strasse 295, 45659 Recklinghausen, Germany; l.prix@ladr.de

**Keywords:** GBM, newcastle disease virus, modulated electrohyperthermia, dendritic cell vaccination, clinical trial, individualized multimodal immunotherapy

## Abstract

**Simple Summary:**

Although multiple meta-analyses on active specific immunotherapy treatment for glioblastoma multiforme (GBM) have demonstrated a significant prolongation of overall survival, no single research group has succeeded in demonstrating the efficacy of this type of treatment in a prospective, double-blind, placebo-controlled, randomized clinical trial. In this paper, we explain how the complexity of the tumor biology and tumor–host interactions make proper stratification of a control group impossible. The individualized characteristics of advanced therapy medicinal products for immunotherapy contribute to heterogeneity within an experimental group. The dynamics of each tumor and in each patient aggravate comparative stable patient groups. Finally, combinations of immunotherapy strategies should be integrated with first-line treatment. We illustrate the complexity of a combined first-line treatment with individualized multimodal immunotherapy in a group of 70 adults with GBM and demonstrate that the integration of immunogenic cell death treatment within maintenance chemotherapy followed by dendritic cell vaccines and maintenance immunotherapy might provide a step towards improving the overall survival rate of GBM patients.

**Abstract:**

Immunotherapies represent a promising strategy for glioblastoma multiforme (GBM) treatment. Different immunotherapies include the use of checkpoint inhibitors, adoptive cell therapies such as chimeric antigen receptor (CAR) T cells, and vaccines such as dendritic cell vaccines. Antibodies have also been used as toxin or radioactive particle delivery vehicles to eliminate target cells in the treatment of GBM. Oncolytic viral therapy and other immunogenic cell death-inducing treatments bridge the antitumor strategy with immunization and installation of immune control over the disease. These strategies should be included in the standard treatment protocol for GBM. Some immunotherapies are individualized in terms of the medicinal product, the immune target, and the immune tumor–host contact. Current individualized immunotherapy strategies focus on combinations of approaches. Standardization appears to be impossible in the face of complex controlled trial designs. To define appropriate control groups, stratification according to the Recursive Partitioning Analysis classification, MGMT promotor methylation, epigenetic GBM sub-typing, tumor microenvironment, systemic immune functioning before and after radiochemotherapy, and the need for/type of symptom-relieving drugs is required. Moreover, maintenance of a fixed treatment protocol for a dynamic, deadly cancer disease in a permanently changing tumor–host immune context might be inappropriate. This complexity is illustrated using our own data on individualized multimodal immunotherapies for GBM. Individualized medicines, including multimodal immunotherapies, are a rational and optimal yet also flexible approach to induce long-term tumor control. However, innovative methods are needed to assess the efficacy of complex individualized treatments and implement them more quickly into the general health system.

## 1. Introduction

Cancer is the second leading cause of death, accounting for about 1 in 6 human deaths. Worldwide, in 2018, about 9.6 million deaths were due to cancer [1]. Between 2013 and 2017, the cancer death rate (mortality rate) in the US was 158/100,000 individuals per year. The rate of new cases (incidence) in a similar period was 442/100,000 individuals per year [2]. Intensive preclinical and clinical research is being performed to find solutions. In some domains, like pediatric hemato-oncology, major progress has been realized towards a cure through systematic randomized controlled clinical trials (RCTs). In each trial, a new experimental arm is assessed versus the best current treatment as the control arm [3,4], in combination with careful monitoring of (long-term) side effects [5].

Despite being an orphan disease, brain tumors are the leading cause of cancer death in males aged 20 to 39 years and the fourth most common cause of cancer death in females in the same age range [6]. Glioblastoma Multiforme (GBM) is the most frequently diagnosed malignant brain cancer in adults and has the worst prognosis [7,8]. The cause of GBM formation is not known. Ageing that progressively suppresses normal immune surveillance has been mentioned to contribute to GBM cell initiation and/or outgrowth [9]. Irradiation is certainly a cause for GBM formation, and the prognosis of a second malignant GBM is extremely poor [10]. Long-term exposure to higher doses of non-ionising irradiation has been associated with the formation of GBM [11]. Finally, viral infections like CMV (variants) have been mentioned as being potential triggers for GBM formation [12]. The classic pillars of treatment for GBM nowadays are neurosurgery, radiochemotherapy, and maintenance chemotherapy [13,14]. In recent years, the standard of care has not changed. Intensive research in different domains has been performed to improve the prognosis of GBM patients, including research in tumor-treating fields, anti-angiogenic treatments, targeted therapies, oncolytic virus therapy, and immunotherapies. The latter term covers different approaches, like restorative immunotherapy, modulating immunotherapy, passive immunotherapy, adoptive immunotherapy, and active-specific immunotherapy with vaccines [15]. For the development and production of mostly personalized cell-based therapies, Good Manufacturing Practice (GMP) facilities are required.

On the occasion of a regulatory audit in July 2019, in connection with the installation of a new GMP facility at the Immune Oncologic Centre in Köln (IOZK, www.iozk.de) and the running GMP-compatible production of IO-Vac^®^ Dendritic cell (DC) vaccines, a discussion was raised with respect to RCTs within the spectrum of delivered individualized multimodal immunotherapy (IMI) activities. The background of this question is a subject of current global debate regarding the future use of controlled RCTs to obtain evidence of the efficacy of immunotherapies. This was exemplified by the symposium organized in Brussels on 22 April 2020, entitled “are randomized trials obsolete?” [16]. On 27 May 2015, the IOZK received a certificate of GMP manufacturing compliance (DE_NW_04_GMP_2015_0030) and approval to produce specific autologous anti-tumor DC vaccines for intradermal injection (DE_NW_04_MIA_2015_0033). Since then, multimodal immunotherapy has been implemented in several domains of cancer. Both certificates were renewed on 20 May 2020 (i.e., DE-NW-04-GMP-2020-0054 and DE-NW-04-MIA-2020-0017).

The IOZK is a translational immune-oncology center specializing in the fast translation of emerging novel insights derived from multiple domains of immunotherapy into clinical applications for use on a compassionate basis (“Individueller Heilversuch”) for patients with cancer. The key medicinal product is IO-Vac^®^, which is an approved Advanced Therapy Medicinal Product (ATMP). The IO-Vac^®^ vaccine consists of autologous mature DCs loaded with autologous tumor antigens and matured with danger signals including the Newcastle Disease Virus (NDV). Over the years, the IOZK has established its value for the treatment of cancer patients. Several case reports and retrospective analyses of patient groups have been published [17,18,19,20]. For the current retrospective analysis, the database was fixed at 28 June 2020, including all records registered from 1 June 2015 to 31 May 2020. Over this 5-year time period, 1456 medical records were initiated at the IOZK. The patients came from 69 countries (with 46% from Germany). From this group of patients, 1098 patients agreed to go through an immune-oncologic evaluation and immunodiagnostic blood sampling in order to study the cell numbers and functioning of their immune compartment, the tumor–host immune interactions, their general health status, and their infection status. Ultimately, 651 were able to, and individually consented to, starting multimodal immunotherapy, after being extensively informed about all aspects of the treatment. These patients belonged to all categories of cancer disease. The three most frequent cancer disease categories were neuro-oncology (42%), digestive oncology (18%), and breast cancer (11%). The domain of neuro-oncology represents almost half of all patients effectively treated at the IOZK. This group of patients is still a very heterogeneous group, including multiple types of brain cancer disease, and including patients at different stages of disease. From the 276 patients, 171 patients (62%) were recorded as having GBM.

Based on this large number of patients and the presence of extensive preclinical, translational, and clinical expertise in immunotherapy for GBM, the discussion about the challenges in setting up RCTs for immunotherapy herein will be focused on IMI for GBM, in particular to demonstrate why RCTs to prove the efficacy of this type of treatment are lacking, despite several meta-analyses pointing to a significant shift in overall survival (OS) rates due to the use of active specific immunotherapy with DC vaccines [21,22,23,24,25]. We aim to discuss this complex problem by reviewing the literature (Section 2, Section 3, Section 4, Section 5, Section 6 and Section 7). Afterwards, we illustrate elements from this narrative review with our own data obtained by a retrospective analysis of our patient records (Section 8, Section 9 and Section 10).

## 2. Current Anti-Cancer Treatment Strategies for GBM

GBM is one of the leading causes of death due to cancer in humans and is a major burden for the community [26,27]. Earlier, in the period of poor imaging possibilities, neurosurgery was the only treatment for GBM, and this was mostly performed to make pathological diagnoses and to temporarily relieve symptoms. Only during the last century other treatment modalities became available, of which radiotherapy was the first approach. Radiotherapy became part of the standard care for GBM in the 1940s. The authors could not find any RCTs from that period. Evidence of the efficacy of radiotherapy was certainly created by the demonstration of a dose–response relationship [28]. An RCT, including radiotherapy, chemotherapy, or best available care for anaplastic astrocytoma, was realized around the same period, clearly demonstrating the effect of radiotherapy on OS [29]. Radiotherapy was shown to improve the median OS of GBM patients by some months, and 60 Gy appeared to be the most efficacious and safe dose [30]. More recently, chemotherapeutic agents and combination treatments have been implemented. The addition of temozolomide (TMZ) during radiotherapy, followed by TMZ maintenance chemotherapy (TMZm), further improved the median OS (again, by some months). Proof of evidence was generated through a prospective RCT. In this trial, patients were stratified by World Health Organization (WHO) performance status, type of surgery, and institution [13]. Later on, the data were presented according to an adapted Recursive Partitioning Analysis (RPA) classification, which included age, WHO performance status, extent of surgery, and mental status as variables, resulting in Class III, Class IV, and Class V patients [14].

The MGMT promotor methylation status of the tumor was rapidly recognized as a key factor in the efficacy of TMZ [31]. Retrospective analyses of available MGMT promotor methylation data in relation to the survival data in the RCT have shed new light on such data [14]. The progression-free survival (PFS) benefit attained through the addition of TMZ to surgery and radiotherapy lost its significance in MGMT promotor unmethylated patients, and the gain in median OS was only 0.8 months (i.e., 24 to 25 days), albeit still being significant. Although the results of the original prospective RCT were published 15 years ago, this so-called Stupp regimen is still the standard of care world-wide for both MGMT promotor-methylated and unmethylated patients. Due to its major effect, even in multivariate analyses, the MGMT promotor methylation status has become part of the in/exclusion criteria for RCTs or is used in the stratification of the randomization. Step by step, more insight into the transcriptomic and genomic dimensions were found, and GBM molecular stratification appeared, pointing to EGFR, NF1, and PDGFRA/IDH1 as playing roles in triggering intracellular pathways but also in influencing the response to anti-GBM treatments, formation of the tumor microenvironment, and tumor spread [32]. A further molecular biological analysis including epigenetic profiling unraveled at least six sub-types of GBM, all having different disease characteristics and prognoses [33].

Radiotherapy and chemotherapy treatment modalities are directed against the cancer itself, but cause acute and long-term side effects to the body. These treatments might, however, also have important effects on the immune system. It is already well-known that neurosurgical removal of the tumor transiently “relieves” the systemic immune system from tumor-induced immunosuppression [34], which returns upon disease progression [35]. Both radiotherapy and TMZ have effects on inflammation and the systemic immune compartment but may also influence the tumor microenvironment [36,37,38,39,40].

The improvement in therapeutic approaches has been paralleled by improvements in imaging technologies. Furthermore, knowledge of the molecular biology domain has increased rapidly, thereby introducing new approaches to the development of targeted treatments with less toxicity. A whole series of targeted therapies have been investigated for GBM, as single drugs in relapsed patients or as add-ons to the Stupp regimen [41,42,43,44,45,46]. Fast molecular biological diagnostic procedures and initiation of adapted drug combinations have opened the door to so-called personalized medicine. For the first time, a particular tumor entity is no longer treated according to pre-designed treatments or study protocols but may become adapted to the individual tumor biology profile of each individual patient. Along the same line, targeted therapies can be implemented to treat any type of cancer, regardless of where in the body it started or the type of tissue from which it developed, pointing to the term “tumor-agnostic treatment” [47].

## 3. Challenges for RCTs

RCTs, originally conceived as a study concept by Sir Austin Bradford Hill (who also developed the criteria for determining a causal association) and applied for the first time in 1948 [48], aim to control for selection bias and allocation bias by balancing patient groups based on known and unknown prognostic factors. Blinding further reduces experimenter and subject biases. The study methodology allows the efficacy of an intervention under investigation to be demonstrated with the greatest amount of evidence in a defined patient population compared to a control population without this particular intervention. RCTs are the gold standard for proving the efficacy of an intervention. Nevertheless, over the years, research has neglected the additional value of observational studies, demonstrating the effectiveness of interventions [49,50,51,52,53,54].

Evolution towards the personalization of medicine is challenging for classic RCTs, especially when related to GBM [55]. Stratification is a requirement to avoid differences between patients treated in an experimental arm versus patients treated in a control arm. The need for larger patient groups has resulted in longer recruitment periods and more expensive clinical trials. The rapid introduction of new drugs has forced clinical researchers to form innovative statistical designs for clinical trials, such as Continual Reassessment Method designs, Sequential Multiple Assignment Randomized Trial designs, and Multi-Arm Multi-Stage clinical designs [56].

At present, most (if not all) single-drug-targeted therapy trials for GBM have failed [57], and a new challenge has emerged. Indeed, GBM is a mixture of different tumor cell clones, including glioma cancer stem cells, and there are dynamic changes in emerging and disappearing clones during the course of the disease and in relation to the treatment(s) given [58,59]. Keeping a patient under a fixed treatment protocol over time, eventually in clinical trials, to treat a dynamically and rapidly changing deadly tumor might no longer be considered appropriate in light of advances in modern medicine. Much more effort should be put into studies on liquid biopsies in patients with GBM [60]. These tests should be performed repeatedly during treatment in order to monitor changes in tumor biology during treatment, as has been shown for other cancer diseases [61].

Inflammatory reactions occurring in the context of immunotherapy might necessitate medical intervention, such as the use of steroids or Bevacizumab, for a short period of time. Due to the deadly nature of the disease, changes over time should be taken into account when treating the patient using a study protocol. Such changes during the protocol might induce drop-out of the trial, which is—from a medical perspective—an emotional burden for the patient. Otherwise, changes in treatment have to be taken as data within the protocol, which creates statistical conflicts with the control group of patients, who must then be treated differently.

One of the particular statistical challenges in demonstrating the efficacy of an active specific immunotherapy is the need to assess the increase of percentage long-term OS, rather than the shift of PFS [62], the latter being the usual primary read-out for testing new drugs in late stage phase II RCTs. A phenomenon of GBM pseudo-progression due to immunotherapy makes the read-out of PFS virtually impossible. Similar to the change in using the MacDonald’s criteria [63] to the RANO criteria [64] to define progression at a time when TMZ was implemented in routine clinical treatment, iRANO criteria have been developed to assess the progression of GBM disease in the context of immunotherapy [65,66]. The increase in the percentage of long-term OS due to DC vaccines as a primary read-out necessitates the use of a much higher number of patients in a RCT, as compared with examining a shift in the median PFS, to yield results showing a significant difference with high enough power. Still, long-term OS is the only relevant read-out from immunotherapy trials.

## 4. Immunotherapy for GBM

Immunotherapy was called a “break-through for cancer” by Science in 2013. In 2011 and 2018, Nobel Prizes for Medicine were devoted to researchers who discovered important insights into the use of immunotherapy for cancer. At present, immunotherapy is definitively an important pillar in anticancer treatment in general, including GBM. Immunotherapy is a very broad term, comprising several technologies. Immunomodulation with checkpoint inhibitors is considered to belong to the domain of tumor-agnostic treatments [47]. Immunotherapies generally do not point to the molecular biological characteristics inside tumor cells, but, rather, to the surface of tumor cells, including a heterogeneous number of known and unknown tumor antigens [67] and immune costimulatory and inhibitory molecules on the surface [68,69,70], as well as the production of cytokines and chemokines that influence the immune system and the inflammatory response [71]. Terms like “cold” versus “hot” tumors, describing differences in lymphocyte infiltration in the tumor microenvironment, have become highly relevant [72]. The tumor mutational burden might be related to the presence of more or less tumor antigens on the surface [73,74]. Some genetic abnormalities, such as the p53 mutation, affect the extent of the expression of MHC molecules on the tumor cell surface [75]. An immunogram is created using the tumor foreignness, general immune status, immune cell infiltration, absence/presence of checkpoints, absence/presence of soluble inhibitors, absence/presence of inhibitory tumor metabolism, and sensitivity of the tumor to immune effectors [76]. The recognition that almost half of GBM tumors consist of cells belonging to the myeloid compartment (e.g., M1/M2 macrophages, tumor-associated macrophages, myeloid-derived suppressor cells, microglia) makes the understanding of the response to treatment and the ultimate outcome of patients more difficult [77,78]. Overall, the tumor microenvironment can be categorized into three functional sub-types that play significant roles in the outcomes of patients [79]. Interestingly, Hallaert et al. [80] also observed that the connection of GBM to the sub-ventricular zone appears to be a novel, independent risk factor leading to a worse prognosis. An association between sub-ventricular zone contact and markers related to the epithelial–mesenchymal transition was discovered [81]. Moreover, there is a definite influence of the tumor and the response to treatment on the systemic immune and inflammatory compartments [34]. Finally, steroids, radiotherapy, and chemotherapy all influence antitumor immune responses [39]; this, again, differs from patient to patient. A strong correlation was found between OS and the systemic immune profile after neurosurgical resection and after radiochemotherapy when the extent of resection and timing of immunotherapy were taken into account [40].

## 5. Risk Factors and Levels of Personalized Medicine

All in all, in addition to the clinical risk profile, the intracellular molecular biology, and the epigenetic profiling of tumor cells, the anatomic location of the tumor, tumor–host interactions in the tumor microenvironment, the systemic immune system, the combination treatment design and the reaction of the body, and dynamic changes in the tumor should be taken into account and used as mandatory stratification tools in the design of RCTs with experimental versus control groups (Table 1). Some of these have been included in novel molecular RPA classification (GBM-molRPA) systems [82].

Immunotherapy covers many treatment modalities. Some of these treatment modalities, such as cytokines, are not personalized to each patient. Other drugs, like antibodies, are themselves not personalized, but are directed against personalized targets on the tumor cell surface. For adoptive cell therapies with chimeric antigen receptor (CAR) T-cells or T cell receptor-transduced T-cells, both the ATMP itself and the target are personalized [83,84,85]. In the domain of active specific immunotherapy with vaccines, the vaccine itself can be an individual product at the level of the antigen carrier (like autologous DCs) and/or at the level of the antigen, such as tumor-derived antigens and, especially, highly individualized tumor-specific epitopes [86,87,88]. Extensive individualization of the treatment drug and target makes the design of a homogeneous experimental arm versus control arm very challenging. By introducing immunotherapy into a first-line treatment combination, new dimensions of personalization for the treatment of GBM have become clear (Table 2).

Overwhelming evidence of the efficacy of DC vaccination for GBM has been presented in preclinical models [89,90,91,92,93,94,95,96,97,98,99,100,101], as well as in phase I and early phase II non-controlled clinical trials [88,102,103,104,105,106,107,108,109,110,111,112,113,114,115,116,117,118,119,120,121,122,123,124,125,126,127,128,129,130,131,132,133,134,135,136,137,138,139,140,141,142,143,144,145,146,147,148,149,150,151,152,153,154,155,156,157,158,159,160,161,162,163,164]. The observation of long-term OS in patients after DC vaccination, noting that the intervention abruptly changes the 100% lethality of disease, has generated strong evidence of its efficacy. Several meta-analyses have demonstrated a significant improvement in long-term OS when GBM patients were treated with DC vaccines as compared with patients given a control treatment [21,22,23,24,25]. Of note, the latter meta-analysis also demonstrated a lack of additional toxicity due to immunotherapy in comparison with the standard of care. This means that the set-up of new phase I or phase IIa clinical trials is no longer innovative.

As has already been pointed out for the changes in molecular sub-clones over time during treatment, evidence that tumor–host immune interactions can also change over time has been obtained. Examples are the downregulation of specific target antigens [165] or the upregulation of PDL1 on tumor cells, allowing them to escape from immune attack during immunotherapy [19]. Dynamic changes in the expression of tumor-associated antigens and immunosuppressive factors in the tumor microenvironment during treatment have also been described [164].

## 6. Current Landscape

On 12 October 2020, we performed a search of Clinicaltrials.gov using the terms “dendritic cell”, “interventional studies” “glioblastoma”, and “interventional phase 2 phase 3 phase 4”. Twenty-nine studies were available, with 12 of them reporting to be recruiting (Table 3). Seven of these studies were RCTs, with six having OS in the read-out analysis. Only one study (NCT04277221) was a phase 3 study. NCT03395587, NCT04115761, and NCT01567202 integrated DC vaccination into the primary standard of care for IDH1wt GBM. Similarly, NCT03548571 integrated DC vaccination into the primary standard of care, but only for MGMT promotor non-methylated patients. The RCTs NCT02465268 and NCT03688178 integrated DC vaccination into the standard of care, but did not mention the requirement of IDH1 wild-type status.

Hence, at the moment of writing, seven RCTs are recruiting and studying DC vaccination for patients with initial diagnosis of GBM. None of these studies mentioned further stratification. The median number of patients used for randomization was 106, ranging from 24 to 136. Nevertheless, based on the analysis above, stratification at multiple levels (as summarized in Table 1) should be used in order to design a proper control group. Such stratification affects the number of patients recruited, and this number seems, in all running trials, to be too small to yield a significant difference with sufficient power in the efficacy of DC vaccination to increase the percentage of patients with long-term OS versus an appropriate control group.

## 7. Non-Scientific Challenges

In the design of larger RCTs with a control arm, clinical researchers have run into new problems. GBM is a deadly disease. The first interest of the patient is the prolongation of a good-quality life. As DC vaccination has already been shown to significantly prolong long-term OS with a good quality of life, it eventually became unethical to prevent patients in the control arm from being given this innovative fourth anti-GBM immunotherapy modality. Therefore, some RCTs published with PFS as the primary endpoint used a cross-over protocol from the control group to the DC vaccination group, either structurally within the trial design (EudraCT 2009-018228-14) [151] or after reaching the primary end-point (NCT00045968) [160]. By doing so, the OS assessment was, in fact, no longer controlled. In the former study, stratification of RPA classification was used. In the latter study, stratification for MGMT methylation was used.

It is notable that the three largest RCTs initiated to date have faced huge logistic/financial problems, as follows: (1) The EudraCT 2009-018228-14 trial aimed to include 146 study objects but was abruptly closed by the sponsor after the inclusion of 135 patients. The reasons for this were kept confidential. Nevertheless, the scientific data were made available, were extremely well-documented, and were partially transferred onto the platform of the EU Project Computational Horizons in Cancer (www.chic-vph.eu). Data on 132 patients were finally brought into a new study, named the Glioma Translat Study, and published [166,167]; (2) NCT00045968 was designed to include 348 study objects with PFS as the primary read-out and OS as the secondary read-out. Eighty sites in four countries were used for recruitment. The study screened 1268 patients and ended up with 331 randomizations into the main study [160]. Patient recruitment was initiated in 2007 but was paused from 2009 to 2011 for economic reasons. The final patient was enrolled in November 2015. This resulted in a recruitment period of nine years, an effective inclusion period of six years, and a mean of four patients per recruiting site at an effective rate of less than one patient per year per site. Five years later, at the time of writing, the final results regarding the randomized data collected on these patients and the answer to the primary question (PFS) have not yet been published; (3) NCT02546102 suspended further patient randomization due to financial reasons [168]. It is not clear how many patients had already been recruited into the trial.

Finally, patients have started searching on blogs and in social media communities for solutions to surviving GBM. An impressive list of complementary medicines and diets is available and freely accessible to patients. Some of these complementary medicines surely influence the course of the disease or the reactivity of the host [169,170,171,172,173]. It seems impossible to control for this when setting up and running RCTs at present.

## 8. Individualized Multimodal Immunotherapy for GBM

Two conclusions have become clear: (1) Individualized treatment at multiple levels, with strategies that follow the dynamic changes in both the tumor and host (e.g., immunity, inflammation) during treatment, is needed for the treatment of GBM; and (2) IMI is becoming part of the first-line treatment for GBM, integrated into the standard of care. The potential inherent immunization component of standard anti-GBM strategies (e.g., neurosurgery, radiotherapy, and chemotherapy) and the proven induction of immunogenic cell death (ICD) through the use of innovative anti-cancer treatments (e.g., modulated electrohyperthermia, oncolytic virus therapy) [174,175] should be exploited and strengthened with active immunization strategies and further optimized with immunomodulatory strategies in the context of optimized complementary medicine. Only broad and long-term immune control over GBM is able to induce long-term OS. The rationale of such a strategy has been published [176]. In Section 8, Section 9 and Section 10, we aim to illustrate several elements discussed in the former narrative review with observational data obtained by a retrospective analysis of our own patient records. Over the years, we have developed a multistep treatment approach, in which ICD is induced by the combination of bolus injections with NDV and modulated electrohyperthermia (mEHT) integrated as an add-in treatment during the alkylating mode of tumor cell killing induced by TMZm chemotherapy, followed by IO-Vac^®^ DC vaccination, in order to actively induce an antitumor immune response in combination with individually adapted immunomodulatory strategies. Finally, ICD treatment is maintained with immunomodulation [19].

Of the 171 recorded patients with GBM in the IOZK database, 90 patients (53%) received IMI in combination with first-line treatment for compassionate use (“individueller Heilversuch”). For the current retrospective analysis, we excluded all patients below 18 and above 75 years of age, patients with a proven pre-history of low-grade glioma and/or proven IDH1 mutation, patients with proven histone mutations, and patients with radiotherapy-induced GBM as a second malignancy, leaving 70 patients for analysis.

The final patient group consisted of 70 patients (33 females, 37 males). The median age of the patients was 50 years (range 18–69 years). The median Karnofsky performance index score was 90 (range 50–100). Clinical patient characteristics are shown in Figure 1. The patients presented for immune diagnostic blood sampling at IOZK at a median of three months after operation (range 0.5–23 months); hence, most sampling was done after radiochemotherapy. General blood counts and immune cell counts and functioning, as analyzed in the routine clinical laboratory, are shown in Figure 2A. A large proportion of patients were lymphopenic at that time—caused by the radiochemotherapy—a finding compatible with published data [39].

The MGMT promotor methylation status of the resected tumor was documented in 52 cases, from which 32 patients were categorized as unmethylated. In 67 patients, the presence of circulating tumor cells (CTCs) was investigated at the time of immune diagnostic blood sampling. Blood samples were sent to Biofocus (https://www.biofocus.de/) for analysis. In 28 patients, no CTCs were detected. CTCs were detected in 39 patients, based on the presence of large cells with mRNA expression for specific oncogenes (EGFR, ERBB2, C-kit, and Telomerase) above a cut-off in comparison with house-keeping mRNA expression. In nine patients, mRNA expression for PDL1 was above the cut-off value of 2 (Figure 2B). In 11 patients, increased mRNA expression for MGMT was observed, of which seven patients were histologically classified as being MGMT promotor unmethylated, while three patients were classified as being methylated (one patient had an unknown histology result). For 16 patients who had CTC without increased mRNA for MGMT, six samples matched with a histologic classification as MGMT promotor methylated, while nine samples mismatched with the histologic classification and were classified as MGMT promotor unmethylated. It should be noted that the categories for both histology-based MGMT promotor methylation and the mRNA expression of MGMT in CTCs were derived from continuous variables linked with methylated versus unmethylated status. Moreover, GBM might depict heterogeneity for MGMT promotor methylation, such that histology sampling errors cannot be excluded. Finally, the presence of CTCs was investigated at a median of three months after operation, thereby potentially illustrating clonal evolution during the first period of radiochemotherapy.

To further illustrate the biological evolutions that occurred during treatment, we looked at the evolution of mRNA expressions in CTCs for the oncogenes EGFR, ERBB2, C-kit, and Telomerase, as well as the mRNA for MGMT and PDL1 over time. For this, available data on mRNA expression in CTCs were sampled and put on a time line for each patient starting from neurosurgery. As mentioned earlier, the patients started immunotherapy at different time points and were treated individually. This is reflected by the use of an individual monitoring schedule for each patient. Figure 3A shows the evolution of the EGFR mRNA expression over time for individual patients. Changes in mRNA expression were only observed in three patients. On the contrary, as shown in Figure 3B, mRNA expression for ERBB2, C-kit, Telomerase, MGMT, and PDL1, in comparison to mRNA expression for the house-keeping gene NADPH, clearly changed over time and was often visible around the appearance of relapse. As an example, the mRNA expression for PDL1 in the CTCs of patients 22,731 and 23,346 was low at the start of immunotherapy treatment. These values, however, increased during treatment and were ultimately accompanied by relapse. Patient 23,346 was treated with pembrolizumab. Patient 24,005 also had a very high level of mRNA expression for PDL1 when the CTCs became positive for the first time. This patient also received four doses of pembrolizumab, after which the CTCs became negative for the PDL1 marker. These descriptive data indirectly demonstrate the possible relative changes in sub-clones during the disease process, which might contribute to the occurrence of relapse and which might be monitored for timely intervention. GBM is thus a dynamic tumoral process. The descriptive analysis illustrates the heterogeneity within these 70 patients at the levels of clinical risk factors, immune variables in the peripheral blood, and molecular tumor biology data, as well as the dynamic processes occurring in the molecular biology of the tumor.

The patients received IMI within the first-line treatment. Table 4 shows the details of the IMI treatment. A median of two (range 0–5) IO-Vac^®^ DC vaccines, 25 (range 0–77) NDV administrations, and 30 (range 0–77) sessions of modulated electrohyperthermia were given to each patient. One IO-Vac^®^ DC vaccine consisted of a median of 12.2 × 10^6^ autologous mature IO-Vac^®^ DCs loaded with autologous tumor proteins derived from tumor lysate (when available and appropriate to the GMP requirements) or obtained as ICD-therapy-induced, serum-derived, antigenic, extracellular microvesicles, and apoptotic bodies.

## 9. A Case of Complete Remission and Specific T-Cell Response

In our retrospective analysis of these 70 patients, one particular innovative finding emerged. As the tumor-specific antigens were not known when using tumor lysate or serum-derived, extracellular vesicles and apoptotic bodies, the immune monitoring in our clinical setting was not a major focus. Nevertheless, proof of principle of ICD therapy in combination with TMZ maintenance chemotherapy followed by vaccination with IO-Vac^®^ DC vaccines loaded with ICD therapy-induced, serum-derived, antigenic, extracellular microvesicles, and apoptotic bodies [19] for the induction of a tumor-neo-epitope-specific immune response was demonstrated in one patient. This 18-year-old patient had an incomplete resection of a left frontal lobe IDH1 wild-type and MGMT unmethylated GBM. The tumor mutational burden was low (0.5 variants/megabase), and there was no evidence for microsatellite instability or germline variants. She was treated with radiochemotherapy and, subsequently, five cycles of TMZm chemotherapy (Figure 4). At presentation, her Karnofsky performance index was 90. She was lymphopenic with 279/uL CD3+ T-cells, 78/uL CD19+ B-cells, and 56/uL CD16+CD56+ NK cells. Her NK cell functioning was below the reference value. She had Th2/Th17 skewing. Her CTCs showed upregulated expression (3.44) of mRNA for MGMT, which was compatible with the known unmethylated MGMT promotor status. The expression of mRNA for PDL1 was 1.13 (cut-off for positivity = 2). She continued treatment for another seven TMZm chemotherapy cycles combined with 5-day ICD treatments, which were given during each TMZm cycle at days 8 to 12. Afterwards, she received two IO-Vac^®^ DC vaccinations loaded with ICD treatment-induced, serum-derived, antigenic, extracellular microvesicles, and apoptotic bodies. Later on, she received two IO-Vac^®^ DC vaccines loaded with tumor-specific peptides based on the individualized tumor-specific neo-antigen detection tests performed at CeGaT (www.CeGaT.de). At the time of writing, she receives monthly maintenance ICD treatments and is still in complete remission. Interestingly, we were able to monitor the tumor antigen-specific T-cell responses as, in this case, the neo-antigens were known. A first sample was available after the fifth TMZm cycle, prior to the addition of multimodal immunotherapy. A second sample was available at the time of blood sampling to prepare the second IO-Vac^®^ DC vaccine (Figure 4). From the data, it is clear that surgery, radiochemotherapy, and five cycles of TMZm did not induce a tumor-specific T-cell response. However, the addition of seven ICD treatments to the last seven TMZm treatments and the first IO-Vac^®^ DC vaccine loaded with ICD-treatment-induced, serum-derived, antigenic, extracellular microvesicles, and apoptotic bodies generated a clear tumor antigen-specific CD4+ and CD8+ T-cell response. The impact of this observation might be meaningful. First, it is not mandatory to have fresh frozen tumor material to prepare a tumor lysate as an antigenic source. This avoids the critical challenges of yielding, freezing, transporting, analyzing, and preparing tumor material within a GMP context. Secondly, and even more importantly, if molecular subclones can change over time, their antigenic profiles can change as well. ICD treatment allows the yield of tumor antigens that are expressed within the body at the time of treatment and makes it possible for the vaccine to immunize against the antigens that are actually present, instead of tumor antigens identified at the time of tumor resection prior to radiotherapy and chemotherapy and hence prior to eventual treatment-induced tumor clone changes.

## 10. Results in Term of OS

As we did not have reference radiology for the independent assessment of new events, given the need to follow the iRANO criteria [65,66] and as OS is certainly the most important outcome in the context of immunotherapy, we focused on the OS of the patients. As described earlier in Section 8, the retrospectively analyzed group of 70 patients was a heterogeneous group of patients. They best reflected adapted RPA class 4 patients, as published by Stupp et al. [14]. This reference was used as a historical control. Data were analyzed using GraphPad Prism version 7.00 for Windows (GraphPad Software, La Jolla, CA, USA, www.graphpad.com). One patient was lost during follow up. The median OS was 20.03 months (Figure 5A) with a 2-year OS of 38.83% (CI95%: +13.04, −13.25) and a 3-year OS of 31.41% (CI95%: +13.35, −12.55). Patients younger than 50 years (21 MGMT promoter unmethylated, 7 methylated, and 7 unknown) had a median OS of 22.07 months, which was not statistically different from the median OS of 18.07 months observed for patients older than 50 years (11 unmethylated, 13 methylated, and 11 unknown). We found MGMT promotor methylation status to be a significant factor (Figure 5B, log-rank test: *p* = 0.0004): patients (*n* = 20) with patients with MGMT promotor methylation having a median OS of 42.85 months versus 11.77 months for patients with unmethylated MGMT promotor status (*n* = 32).

For 16 patients, local therapy (neurosurgery, radiochemotherapy) was followed by IMI (Group 1). ICD therapy (the combination of IV bolus injections of NDV together with mEHT [19]) integrated into the TMZm chemotherapy cycles followed by IO-Vac^®^ DC vaccines and maintenance ICD therapy was given to 46 patients (Group 2). Eight patients started with IMI after the last TMZ maintenance chemotherapy cycle (Group 3). The OS data were significantly different for the three treatment groups: 13.08 months for group 1, 22.46 months for group 2, and undefined for group 3 (median follow-up of surviving patients: 28.59 months, range 26.3–55.9 months; Figure 6A). The latter likely points to favorable selection of a small group of patients who experienced no events until after the end of the maintenance chemotherapy and profited from subsequent IMI treatment. The median OS of the patients with unmethylated MGMT promotor status was 11.25 months for group 1 (*n* = 7) and 18.07 months for group 2 (*n* = 21), with a 2-year OS of 0% in group 1 versus 17.18% (CI95% +31.65, −15.86) in group 2 (Figure 6B, log-rank test: *p* = 0.0273). The median OS of the patients with methylated MGMT promotor status was 42.85 months for group 2 (*n* = 15) with a 2-year OS of 66.66% (CI95%: +19.3, −32.96). There were only three patients in group 1 who had a tumor with MGMT promotor methylation status. Within treatment group 2, the differences in the OS curve due to MGMT promotor methylation status were significant (Figure 6C, Log-rank test: *p* = 0.0036).

Considering these data and the data published by Stupp et al. [14], patients with MGMT promotor unmethylated status have no relevant benefit from the addition of TMZ [14] or from treatment with IMI alone. However, the data from treatment group 2 suggest the potential benefit of the integration of ICD treatment within TMZm chemotherapy followed by IO-Vac^®^ DC vaccinations, which was found to increase the median OS by about six months. On the other hand, treatment with IMI after local therapy for patients with MGMT promotor methylated GBM yielded similar median OS data as treatment with TMZm chemotherapy, while the integration of ICD treatment with TMZm chemotherapy followed by IO-Vac^®^ DC vaccinations increased the median OS further—by about 18 months. These data are worth validating in prospective clinical research.

## 11. Conclusions

In this paper, we reviewed the multiple challenges related to the use of immunotherapy RCTs for GBM (Section 2, Section 3, Section 4, Section 5, Section 6 and Section 7) and illustrated these challenges in a retrospective analysis of GBM patients treated at IOZK (Section 8, Section 9 and Section 10). GBM treatment is affected by the complexity of the tumor, the complexity of the immune and inflammatory micro-environment, the complexity of combined treatment approaches, and the complexity of dynamic changes occurring within the tumor, the tumor microenvironment, and the immune system. The set-up of immunotherapy RCTs for GBM is affected by the need for multiple stratifications to create an appropriate control group. Both stratification and the read-out of OS necessitate the use of a large number of study participants. The costs to run such RCTs are related not only to the Good Clinical Practice (GCP) documentation but also to the production of the ATMP in a GMP environment. As already mentioned, there has been an abundance of smaller phase I and phase II single-arm clinical trials, and several meta-analyses have shown the efficacy and safety of DC vaccination for GBM treatment. The high cost of non-innovative small RCTs with predicted negative outcomes due to a lack of appropriate stratification makes them not appropriate at this time due to limited resources. Models for OS and immunotherapy responses in patients with GBM based on DNA-methylation-driven, gene-based, molecular classifications and multi-omic analyses may be better tools for predicting the efficacy individually for each patient [177]. However, patients suffering today from GBM need better treatments, including the use of first-line immunotherapy to fight their cancers.

The multimodality of immunotherapy, besides the standard of care, presents a further challenge to the classic step-wise research methodology used in clinical research. IOZK aims to contribute to the general knowledge and communicate their gained experiences to the wider scientific community. The ATMP IO-Vac^®^ DC vaccine has been used as part of IMI treatment in the framework of “Individueller Heilversuch”. Each patient is treated at all personalized levels necessary. For each individual patient, the best possible solution for their medical needs is worked out. Prospective medical research questions on groups of patients are not generated. Nevertheless, the IOZK aims to repetitively freeze the database at certain time points and sample patient data to carry out scientifically correct retrospective analyses. The current analysis suggests that the addition of ICD treatment with NDV and mEHT during the application of TMZ maintenance chemotherapy, followed by IO-Vac^®^ DC vaccinations and maintenance ICD treatment, may be beneficial for both MGMT promotor unmethylated and methylated GBM patients.

A final challenge in the broad implementation of innovative therapies like IMI, given that its effectiveness has been accepted without the use of RCTs, is the expected cost versus the length of additional survival. The production of personalized ATMPs under GMP conditions is expensive. The control over the ATMP quality, and hence over the costs, belongs to the authorities, who are also responsible for health policies in general. This topic points to the need for a cost-effectiveness analysis to compare the costs and outcomes of treatment options. The aim is to ensure the greatest possible health benefits are attained with a given budget. For macroeconomic considerations, cost-effectiveness thresholds (CETs) are usually related to the gross domestic product (GDP) per capita [178]. To the best of our knowledge, no representative data related to the integration of IMI into standard of care treatment of GBM are available. Single studies have appraised the additional cost of targeted therapies per year of survival if combined with chemotherapy. In non-small-cell lung cancer, for example, the additional cost for using atezolizumab in combination with carboplatin/nab-paclitaxel amounts to 333,199 USD per quality-adjusted life year [179]. Understandably, the respective costs are difficult to calculate and depend heavily on the type of cancer, the treatment in question, and the parameters used in the cost-effectiveness study. Accordingly, different methods have been described [180]. What we can say is that overall spending on cancer drugs has been increasing. For example, spending on cancer drugs rose from €7.6 billion in 2005 to €19.1 billion in 2014 in the EU [181]. For some types of cancer, calculations are available. Between 2007 and 2012, the mean amounts of money spent in the first year after diagnosis was $35,849, $26,295, $55,597, and $63,063 for breast, prostate, lung, and colorectal cancers, respectively [182]. In view of the overall increase in costs, even higher expenditure must be expected today. A more recent study on breast cancer reported the average cost per patient in the year after diagnosis as being between $60,637 and $134,682, depending on the cancer stage [183]. Newly approved pharmaceuticals easily cost more than $100,000 US per year. Costs are driven by various factors that are not proportional to their often modest additional benefits [184,185], exceeding the cost-effectiveness thresholds. In comparison, the addition of modulated electrohyperthermia to dose-dense temozolomide for the treatment of recurrent GBM has proven to be cost-effective [186]. Taken together, it is methodically difficult to compare the cost–benefit ratio of the integration of IMI into standard of care treatment. Considering the far higher overall costs of targeted therapies, one may expect that the ratio would not be unfavorable.

## Figures and Tables

**Figure 1 cancers-13-00032-f001:**
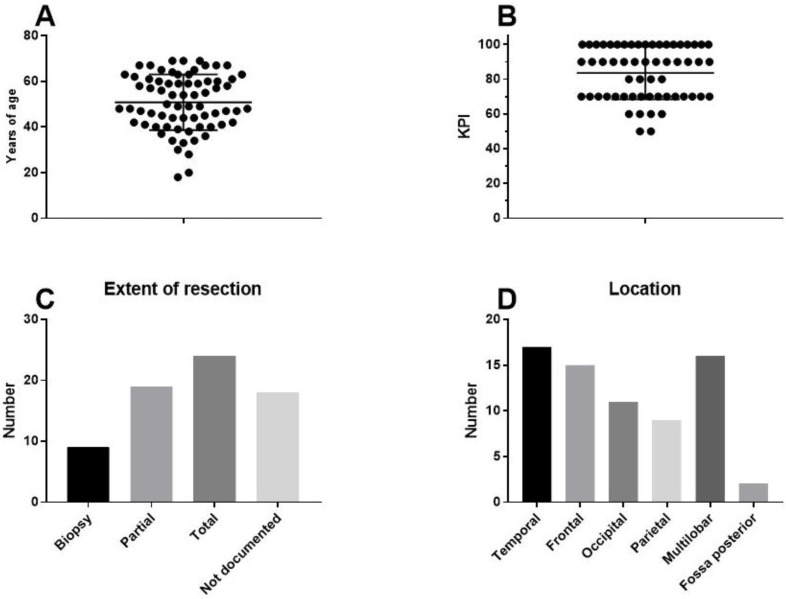
Clinical characteristics of the patients. Seventy adults with primary GBM receiving first-line standard of care in combination with individualized multimodal immunotherapy were included in this retrospective analysis: (**A**) age distribution; (**B**) distribution of Karnofsky performance index scores; (**C**) reported resection (number of patients); and (**D**) reported location (number of patients).

**Figure 2 cancers-13-00032-f002:**
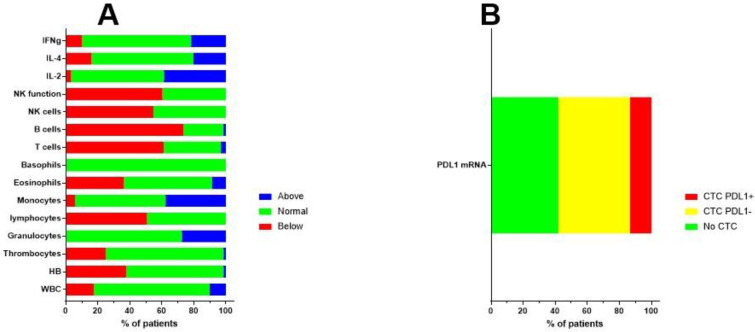
Immune variables before the start of immunotherapy. (**A**) Before the start of individualized multimodal immunotherapy, blood was drawn and sent to the routine clinical lab for analysis. Different immune variables are shown. The percentages of patients with values above (blue), within (normal, green), or below (red) the normal range are shown. (**B**) Blood was also sent to Biofocus in order to detect circulating tumor cells based on mRNA expression of GBM-related oncogenes. When cells were detected, the RNA expression for PDL1 was subsequently analyzed and cells were defined as negative (yellow) or positive (red) for the expression of mRNA for PDL1.

**Figure 3 cancers-13-00032-f003:**
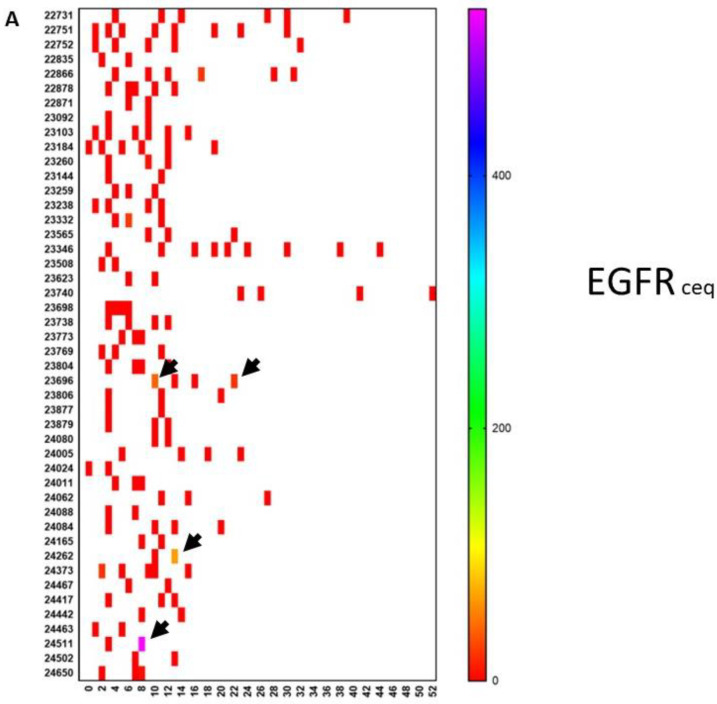
Evolution of oncogene expression in circulating tumor cells. Circulating tumor cells were measured repetitively in 46 patients. Each patient is referenced with a number on the *y*-axis. Time in months is indicated on the *x*-axis. The scale of the color of each test is shown on the right-hand side. (**A**) The level of mRNA expression for EGFR is expressed as the ceq (cell equivalent). Most of the values are negative (red). However, as indicated by the arrows, some patients showed an upregulation of EGFR during the disease course. (**B**) This shows a similar data set-up as in panel A. For each patient, data on mRNA expression are relative to the house-keeping gene GADPH and are shown in a particular color, for which the scale is shown on the right-hand side. For each patient, up to five lines are shown at different moments during the disease course. From top to bottom, values for ERBB2, c-Kit, telomerase, MGMT, and PDL1 are shown for each patient. Stars indicate new events. A † indicates when the patient died. An arrow to the right indicates the time at which the patient was censored in the analysis.

**Figure 4 cancers-13-00032-f004:**
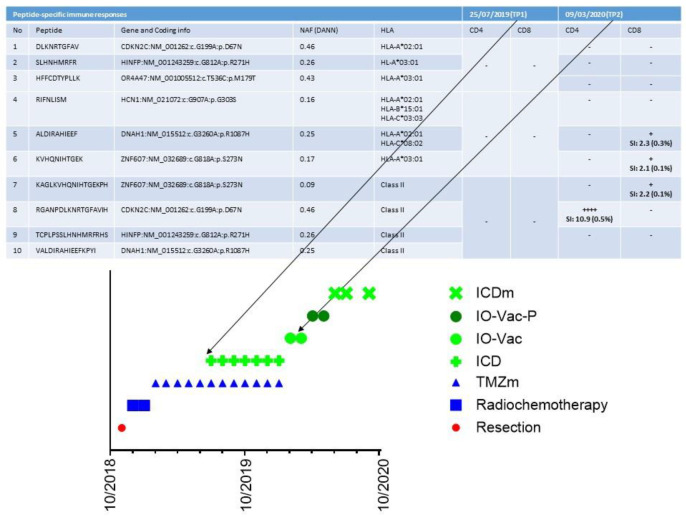
Detection of tumor-specific T-cell clones. The treatment timeline for patient 24442 and its multiple components are shown in the bottom part of the figure. TMZm indicates five days of temozolomide maintenance treatment in cycles (every 28 days). ICD indicates immunogenic cell death (ICD) treatment consisting of the combination of five injections with Newcastle Disease Virus and five sessions of modulated electrohyperthermia. IO-Vac indicates a vaccination cycle including six ICD treatments and an injection of IO-Vac^®^ DC vaccine, consisting of autologous mature dendritic cells loaded with ICD-treatment-induced, serum-derived, antigenic, extracellular microvesicles and apoptotic bodies. IO-Vac-P vaccination cycles are equal to IO-Vac vaccination cycles, but the DCs are loaded with tumor-specific neo-peptides. The upper part of the curve shows the specific peptide sequences, the respective gene and coding information, the Novel Allele Frequency (NAF), and the HLA phenotype. TP1 and TP2 indicate the two respective time points at which T-cells were frozen for immune monitoring purposes. SI is the stimulation index, the ratio of polyfunctional activated CD4+ or CD8+ T-cells (positive for at least two activation markers from CD154, IFN-g, TNF, and/or IL-2) in the peptide-stimulated sample, compared with the unstimulated control. Additionally, the percentage of activated CD4+ or CD8+ T-cells (positive for at least one activation marker of CD154, IFN-g, TNF, and/or IL2) above the background and after in vitro amplification is given. This percentage does not directly reflect the frequencies in vivo. SI ≥ 2: weak response (+); SI ≥ 3: positive response (++); SI > 5: strong response (+++); SI > 10: very strong response (++++). Peptides 1–3, 4–6, and 7–10 were pooled for the analysis of TP1. Peptides 9 and 10 were pooled for the analysis of TP2.

**Figure 5 cancers-13-00032-f005:**
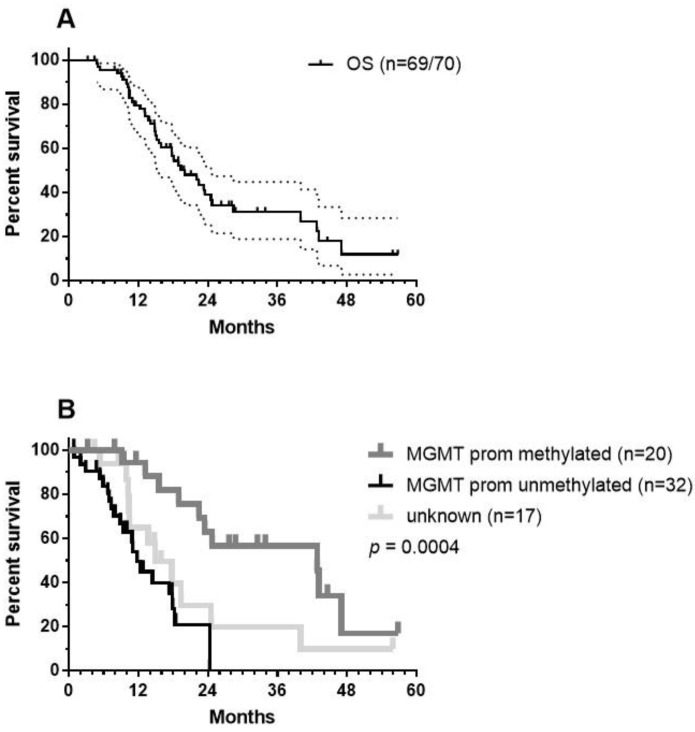
OS data: (**A**) OS data of the total patient group (CI95% values are shown). One patient out of 70 was lost during the follow up period and (**B**) the patient group was divided according to MGMT promotor methylation status—methylated (grey), unmethylated (black), data not registered (light grey). The *p*-values show significance using the Log-rank (Mantel–Cox) test.

**Figure 6 cancers-13-00032-f006:**
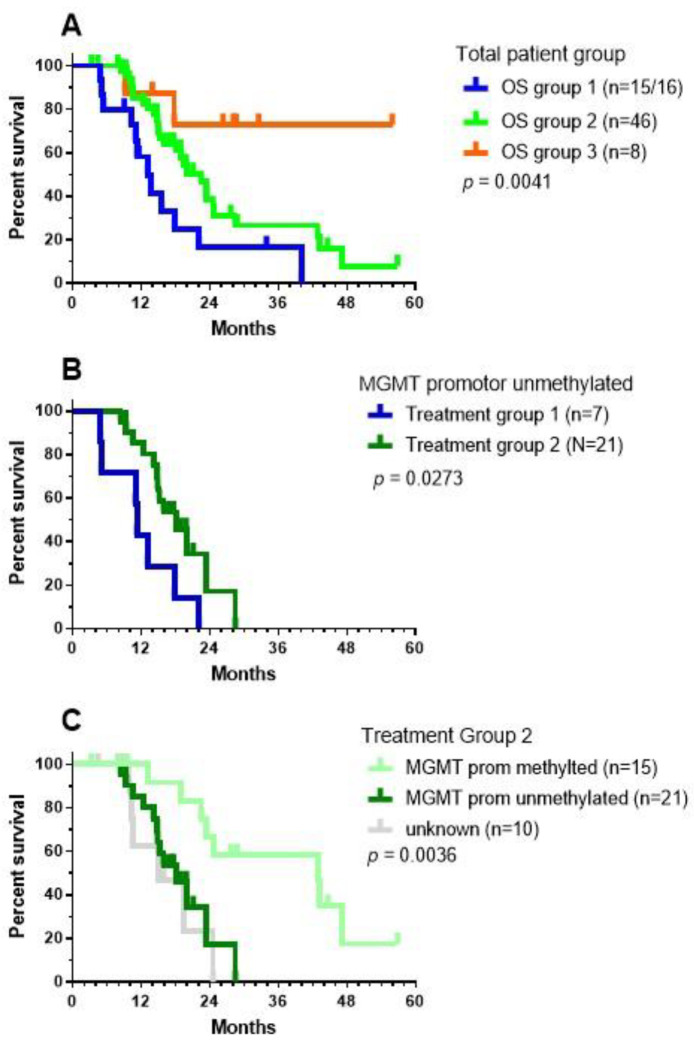
OS data for the treatment groups. As explained in the text, patients were categorized into three different treatment groups: (**A**) OS data for the three different patient groups; (**B**) OS data for the patients without MGMT promotor methylation belonging to treatment groups 1 (blue) and 2 (green); and (**C**) OS data of patients from treatment group 2, divided according to MGMT promotor methylation status. The *p*-values show the significance calculated using the Log-rank (Mantel–Cox) test.

**Table 1 cancers-13-00032-t001:** Risk factors for stratification in randomization or for in/exclusion criteria.

1. Recursive Partitioning Analysis (RPA) clinical classification Grading Extent of resection Age Karnofsky performance index Mental status Dose of radiotherapy 2. Molecular biology of tumor Tumor mutational burden Epigenetic sub-typing Molecular machinery of tumor cell clones Metabolic features of tumor 3. Load of glioma cancer stem cells in connection to periventricular zones of the brain 4. Tumor–host immune reaction Tumor antigen expression Check point expression Inflammatory response, M1/M2 balance, Tumor-associated macrophages, myeloid-derived suppressor cells, microglia reactivity T-cell infiltration Vascularization and oxidative stress 5. Systemic immune compartment Cell numbers under standard of care treatment Th1/Th2 balance Presence of Treg Presence of MDSC Level of natural killer cell reactivity 6. Systemic treatments outside standard of care Steroids Anti-angiogenic drugs Complementary medicines

**Table 2 cancers-13-00032-t002:** Levels of personalized medicine for Glioblastoma Multiforme (GBM).

1. Molecular biology of tumor Tumor mutational burden Epigenetic sub-typing Molecular machinery of tumor cell clones Metabolic features of the tumor 2. Tumor–host immune reaction Tumor antigen expression Check point expression Inflammatory response, M1/M2 balance, TAM, MDSC, microglia reactivity T-cell infiltration Vascularization and oxidative stress Load of glioma cancer stem cells in connection to periventricular zones of the brain 3. Immune reactivity against the tumor Th1/Th2 balance Presence of Treg Presence of MDSC Level of NK cell reactivity 4. Reaction of the immune system upon other treatments Sensitivity to radiotherapy Sensitivity to chemotherapy Use of steroids Use of anti-angiogenic drugs 5. Response to treatment 6. Immunotherapy components Tumor antigens Patient-derived cell products

**Table 3 cancers-13-00032-t003:** ClinicalTrials.gov Search Results on 12/10/2020 for “dendritic cell”, “interventional studies” “glioblastoma”, and “interventional phase 2 phase 3 phase 4”.

Label	Phase	Number of Patients	Randomized	Status	Primary Outcome	Estimated Study Completion
NCT00576537	2	50	No	Completed	Safety/Toxicity	10/2011
NCT02649582	1 + 2	20	No	Recruiting	OS, Safety/Toxicity Feasibility	12/2020
NCT00846456	1 + 2	20	No	Completed	Toxicity, Immune response	02/2013
NCT00323115	2	11	No	Completed	Immune response, Toxicity, PFS	07/2013
NCT02366728	2	100	RCT	Active, not recruiting	OS, DC migration	08/2020
NCT01204684	2	60	RCT	Active, not recruiting	PFS, OS	01/2021
NCT03927222	2	48	No	Recruiting	OS, DC migration	12/2023
NCT01006044	2	26	No	Completed	PFS, Toxicity	08/2014
NCT03395587	2	136	RCT	Recruiting	OS, PFS	06/2023
NCT04523688	2	28	No	Not recruiting	PFS	12/2025
NCT03548571	2 + 3	60	RCT	Recruiting	PFS, OS	05/2023
NCT04115761	2	24	RCT	Recruiting	PSF	06/2022
NCT03014804	2	0	RCT	Withdrawn		
NCT03879512	1 + 2	25	No	Recruiting	OS, PFS	01/2022
NCT02772094	2	50	No	Unknown	OS, Toxicity	12/2016
NCT01567202	2	100	RCT	Recruiting	Response, PFS, OS	02/2020
NCT01213407	2	87	RCT	Completed	PFS, OS	11/2015
NCT01291420	1 + 2	10	No	Unknown	Immune response	Unknown
NCT02465268	2	120	RCT	Recruiting	OS, Immune response, PFS	06/2024
NCT04277221	3	118	RCT	Recruiting	OS, PFS	12/2022
NCT00323115	2	11	No	Completed	T-cell response	07/2013
NCT03400917	2	55	No	Completed	OS	02/2023
NCT02546102	3	414	RCT	Suspended	OS	12/2021
NCT01280552	2	124	RCT	Completed	OS	12/2015
NCT00045968	3	348	RCT	Unknown	PFS	11/2016
NCT03688178	2	112	RCT	Recruiting	OS, Varlimumab safety, Treg level	03/2025
NCT01759810	2 + 3	60	No	Enrolling by invitation	OS	12/2020
NCT02754362	2		No	Withdrawn	Immune response	06/2019
NCT04388033	1 + 2	10	No	Recruiting	Safety, PFS	12/2023

DC: dendritic cell; OS: overall survival; PFS: progression free survival; RCT: randomized controlled trial.

**Table 4 cancers-13-00032-t004:** Treatment details.

	Vaccine	Local Hyperthermia	NDV	DC Total	DC/Vaccine
N	69	70	69	70	112
Minimum	0	0	0	0	2,400,000
25%P	1	13	17	6,950,000	8,075,000
Median	2	25	30	20,115,000	12,200,000
75%	2	42	42	34,200,000	19,145,000

NDV: Newcastle Disease Virus.

## Data Availability

The data presented in Section 8, Section 9 and Section 10 are available on request from the corresponding author. The data are not publicly available due to privacy reasons.

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
