# Peer review of "Randomized Controlled Immunotherapy Clinical Trials for GBM Challenged"

_cancers, 2020, doi:10.3390/cancers13010032_

Round 1
Reviewer 1 Report
This is a very nice comprehensive review about immunotherapy and randomized trials of GBM. This review also provides overview(s) about various immunotherapy approaches with their biological backgrounds. I enjoyed reading this review. There are a few grammatical issues. Other than that I have no major comments.
Author Response
This is a very nice comprehensive review about immunotherapy and randomized trials of GBM. This review also provides overview(s) about various immunotherapy approaches with their biological backgrounds. I enjoyed reading this review. There are a few grammatical issues. Other than that I have no major comments.
ANSWER
We thank the reviewer for these positive comments. Meanwhile the manuscript has undergone extensive English editing via the services provided by Cancers.
Reviewer 2 Report
This article has the following aims:
1) To be a review of the state of the art in immunotherapy trials (sections 1-5), including a systematic review of current trials (section 6, 7).
2) A historical summary of the patients seen in the institution of the authors (section 8), including results as figures.
3) A one-case report with data from one patient in which the therapy was successful (section 9).
4) Survival analysis of a subset of patients from their institution.
Overall, this article has a too ambitious goal and gets losts in its focus. Is this a review? A report of results? If this is the latter, then, the methodology is not described sufficiently.
In my view the main problem is that the article gets lost in providing tons of information while avoiding some critical issues. For example, the patient that survived (section 9), in fact received most of the ICD treatment concomitantly to TMZ, and then right after that, two vaccines that are not clearly described. The patient survived 8 months so far after all these treatments, which is not unexpected, given the fact that she is young.
The graphs in figure 3B are very interesting but totally inintelligible. The only I got from them is that each patient seems to have been studied at particular timepoints, not standardised, very frequently at the beginning (of which time point?) but for not more than 2 months.
Finally, in the survival curves, they analyse a small cohort, based on MGMT status (but not available for a significant part of the patients). At this point of the paper, I was completely lost regardin which were the treatment groups 1, 2 and 3.
In summary, this paper needs to be rethinked and organised. Either split into two, one review and one report, or something else. Note that if results are being reported, the study design and the tests performed need to be clearly stated.
Author Response
This article has the following aims:
1) To be a review of the state of the art in immunotherapy trials (sections 1-5), including a systematic review of current trials (section 6, 7).
2) A historical summary of the patients seen in the institution of the authors (section 8), including results as figures.
3) A one-case report with data from one patient in which the therapy was successful (section 9).
4) Survival analysis of a subset of patients from their institution.
Overall, this article has a too ambitious goal and gets losts in its focus. Is this a review? A report of results? If this is the latter, then, the methodology is not described sufficiently.
ANSWER
We thank the reviewer for this comment. We agree that this total package was indeed the aim of the manuscript. We regret that this is experienced as “tons of information” by this reviewer. The first point (section 2 to 7 of the text) was meant to be a review of the state of the art in immunotherapy trials, which belongs in fact to the special topic “Challenges and Opportunities for effective cancer immunotherapies”. The historical summary of the patients (section 8 of the text), the one-case report (section 9 of the text) and the survival analysis (section 10 of the text) are illustrative to demonstrate elements mentioned in the first point. Of note, the survival analysis (put here in the comment as fourth point) belongs to the historical summary of the patients (mentioned here as the second point). The case report (the third point in this comment) demonstrates one particular technical point related to the introduction of tumor-specific T cell responses using immunogenic cell death therapy in combination with TMZ and using dendritic cells loaded with ICD therapy-induced serum-derived antigenic extracellular microvesicles and apoptotic bodies – a phenomenon which is not described to that extent till now, but having major implications for future approaches when actual tumor antigens are not available via other sources. We have made this now more clear into the text.
(a) Line 93-95: “We aim to discuss this complex problem by reviewing the literature (section 2 to 7). Afterwards we illustrate elements from this narrative review with own data obtained by a retrospective analysis of our patient records (point 8 to 10).”.
(b) Line 334-336: “We now aim to illustrate in section 8 to 10 several elements discussed in the former narrative review with observational data obtained by retrospective analysis of our own patient records.”
(c) Line 474: “In our retrospective analysis of these 70 patients, one particular innovative finding emerged.”.
(d) Line 511: We added “in section 8” to the text in section 10.
(e) Line 559-561: “In this paper, we reviewed the multiple challenges related to immunotherapy RCTs for GBM (section 2 to 7), and illustrated these challenges in a description of a retrospective analysis of GBM patients treated at IOZK (section 8 to 10).”.
We believe that the discussion on the challenges and opportunities of RCTs can be placed in parallel to an illustrative retrospective analysis of our own innovative observational data.
In my view the main problem is that the article gets lost in providing tons of information while avoiding some critical issues. For example, the patient that survived (section 9), in fact received most of the ICD treatment concomitantly to TMZ, and then right after that, two vaccines that are not clearly described. The patient survived 8 months so far after all these treatments, which is not unexpected, given the fact that she is young.
ANSWER
We fully agree with the reviewer that the survival of this patient is not remarkable (yet). As explained above, this patient illustrates a particular technical issue for the yield of antigens for use in the vaccine thereby obtaining a tumor antigen-specific immune response. We have made this clear into the text. Line 471-479: The impact of this observation might be meaningful. First, it is not mandatory to have fresh frozen tumor material for preparing a tumor lysate as antigenic source. This avoids the critical challenges to yield, freeze, transport, analyze and prepare tumor material within a GMP context. Second and even more important, if molecular subclones can change over time, their antigenic profile can change as well. The ICD treatment allows the yield of tumor antigens that are expressed within the body at time of treatment, and makes it possible that the vaccine is immunizing against those antigens that are actually present, instead of tumor antigens taken at time of tumor resection prior to radiotherapy and chemotherapy, and hence prior to eventual treatment-induced tumor clone changes.”.
We have avoided to describe again all the details of the IO-Vac® DC vaccine production. We have added the appropriate reference 3 at several locations throughout the text. We repeated this reference now also in section 9 (Line 475). The details of the production process of IO-Vac is provided in this open-source publication.
The graphs in figure 3B are very interesting but totally inintelligible. The only I got from them is that each patient seems to have been studied at particular timepoints, not standardised, very frequently at the beginning (of which time point?) but for not more than 2 months.
ANSWER
We have clarified this in the text. Line 408-411: “For this, available data on mRNA expression in CTCs were sampled and put for each patient in a time line starting from neurosurgery. As mentioned earlier the patients presented for starting immunotherapy at different time points, and were treated individually. This is reflected by an individual monitoring schedule for each patient.”.
These patients indeed did not belong to a fixed study protocol. They were all treated individually. They presented at different time points for the first time, which was described in line 351 “The patients presented for an immune diagnostic blood sampling at IOZK at a median of 3 months after operation (range 0.5–23 months); hence, mostly after radiochemotherapy.”.
The main message here is that the tumor biology, indirectly reflected by the analysis of these cells in peripheral blood, is changing over time, thereby demonstrating GBM as a dynamic process. We have tried to make our observation and conclusion stronger by adding in line 423: “GBM ermerges as a dynamic tumoral process.”.
Finally, in the survival curves, they analyse a small cohort, based on MGMT status (but not available for a significant part of the patients). At this point of the paper, I was completely lost regarding which were the treatment groups 1, 2 and 3.
ANSWER
This is a retrospective analysis. Not each patient who was referred had data on MGMT promotor methylation. Obviously this test is not done as routine test in several countries. The reality going on during the last 5 years in different countries is reflected in the current retrospective analysis. Nevertheless, for the patients of the total group, and the patients belonging to the largest group (group 2), it remained a significant factor for OS in those patients where the data were available.
The request for clarification of the three patient groups at this stage is correct. We thank the reviewer for bringing this point into attention. We have moved the description of the three treatment groups from the last part of section 8 to section 10 (Line 524-528), so that it becomes clear for the reader what the three groups mean. “For 16 patients, local therapy (neurosurgery, radiochemotherapy) was followed by IMI (Group 1). ICD therapy (the combination of IV bolus injections of NDV together with mEHT {Van Gool, 2018 #8428}) integrated into the TMZm chemotherapy cycles followed by IO-Vac® DC vaccines and maintenance ICD therapy was given in 46 patients (Group 2). Eight patients started with IMI only after intake of the last TMZ maintenance chemotherapy (Group 3).”.
In summary, this paper needs to be rethinked and organised. Either split into two, one review and one report, or something else. Note that if results are being reported, the study design and the tests performed need to be clearly stated.
ANSWER
The concern of the reviewer is well taken, and we have tried to meet to the requests as good as possible, but we believe that the text as such has a strong value given the narrative review aspect and the concrete illustration derived from real life. We have stressed at several places that we describe a retrospective analysis of a group of patients treated individually at the IOZK.
We want to stress at this stage that the methodologies for accurate patient group selection and for survival analysis have been described into detail in the manuscript, and that the methodology for the analysis of the mRNA expression in the CTCs described in section 8, and the immune response data described in section 9, have been referenced to the respective websites of the companies who did the test.
Reviewer 3 Report
The authors have written an interesting study discussing a complicated topic. It is obvious that they would like to get rid of RCT for immunotherapy and GBM. I think it would help the readers if the authors discussed why the use of RCT was started. The authors should discuss the bias that investigators bring to their research and the role of RCT to remove that. IN the 70's multiple studies were presented with "cures" of e very tumor. When repeated in RCT the treatment arm did worse. It would also be helpful to place the cost of immunotherapy in context: how many hundreds of thousands of dollars does each patient cost for what length of survival. This is a very complex question and I applaud the authors for discussing it.
Author Response
The authors have written an interesting study discussing a complicated topic. It is obvious that they would like to get rid of RCT for immunotherapy and GBM. I think it would help the readers if the authors discussed why the use of RCT was started. The authors should discuss the bias that investigators bring to their research and the role of RCT to remove that. In the 70's multiple studies were presented with "cures" of every tumor. When repeated in RCT the treatment arm did worse.
ANSWER
We thank the reviewer for this suggestion. The discussion is not aimed for own purposes but for all clinicians working in this field. All experience the potential of DC therapies to improve OS with good quality of life, but all face similar challenges in further clinical development: the assessment of effectiveness of complex innovative treatments with the “gold standard” RCT being accompanied with reimbursement implications for these complex innovative treatments.
First of all, we have changed the term in the title and in line 56 towards “Randomized controlled clinical trials (RCTs)”, instead of controlled randomized clinical trials. Next we have introduced an introductory paragraph section 3, making the point from the reviewer clear. Line 151-159: “RCTs, originally conceived as study concept by sir Austin Bradford Hill (who also developed the criteria for determining a causal association) and for the first time applied in 1948 [36], aim to control for selection bias and allocation bias, balancing patient groups for known and unknown prognostic factors. Blinding further reduces experimenter and subject biases. The study methodology allows to demonstrate with highest evidence the efficacy of an intervention under investigation in a defined patient population compared to a control population without this particular intervention. RCTs are the gold standard to prove efficacy of intervention. Nevertheless, literature over years appeal not to neglect the additional value of observational studies demonstrating effectiveness of interventions [37-42]..”.
It would also be helpful to place the cost of immunotherapy in context: how many hundreds of thousands of dollars does each patient cost for what length of survival. This is a very complex question and I applaud the authors for discussing it.
ANSWER
We thank the reviewer for rising this point. This is indeed a very difficult and complex issue that goes far beyond our expertise. Nevertheless we researched in the literature, and added a paragraph in the text. Line: 589-617: “A final challenge for broad implementation of innovative therapies like IMI, given its effectiveness is accepted without RCTs, is the expected cost for what length of additional survival. The production of personalized ATMPs under GMP conditions are expensive. The control over the ATMP quality, and hence over the costs, belongs to the authorities, who are also responsible for the health policy in general. This topic points to the cost–effectiveness analysis, a broad socioeconomic field that compares the costs and outcomes of treatment options. The aim is to ensure the greatest possible health benefits with a given budget. For macroeconomic considerations the Cost-effectiveness thresholds (CETs) are usually related to the gross domestic product (GDP) per capita [166]. To the best of our knowledge, no representative data are available related to IMI integrated in the standard of care of GBM. Single studies appraise the additional cost per year of survival of targeted therapies if combined with chemotherapy. In non-small-cell lung cancer as example, the additional costs for using atezolizumab in combination with carboplatin/nab-paclitaxel amount to 333,199 USD per quality-adjusted life year [167]. Understandably, the respective costs are difficult to calculate and depend heavily on the type of cancer, the treatment in question and the parameters used in a cost-effectiveness study. Accordingly, different methods have been described [168]. What we can say is that overall spending on cancer drugs has been increasing. For example, spending on cancer drugs rose from €7.6 billion in 2005 to €19.1 billion in 2014 in the EU [169]. For some types of cancer, calculations are available. Between 2007 and 2012, mean spending was $35,849, $26,295, $55,597, and $63,063 in the first year after diagnosis of breast, prostate, lung, and colorectal cancer, respectively [170]. In view of the overall increase in costs, even much higher expenditure must be expected today. A more recent study on breast cancer reports average costs per patient in the year after diagnosis between $60,637 and $134,682, depending on the stage [171]. Newly approved pharmaceuticals easily cost more than 100.000 USD per year. Costs are driven by various factors that are hardly in any reasonable relation to their often modest additional benefits [172,173], exceeding the cost-effectiveness thresholds. In comparison, modulated electrohyperthermia concurrent to dose-dense temozolomide in the treatment of recurrent GBM has proven to be cost-effective [174]. Taken together, it is methodically hardly feasible to compare the cost-benefit ratio of IMI integrated in standard of care. Considering the far higher overall costs of targeted therapies, one may expect that the ratio is not unfavorable.”.